# Explainable machine learning for predictive modeling of blowing snow detection and meteorological feature assessment using XGBoost-SHAP

Feng Wang◉[1,2]*, Xinrang Wang[2], Sai Li◉[3]

**1** School of Civil Engineering, Beijing Jiaotong University, Beijing, China, **2** Guoneng Shuohuang Railway Development Co., LTD, Yuanping, Shanxi Province, China, **3** School of Civil Engineering, Shijiazhuang Tiedao University, Shijiazhuang, Hebei Province, China

* wf21114126@163.com (FW)

## Abstract

Accurate forecasting of blowing snow events is vital for improving numerical models of snow processes, yet traditional predictive methods often lack interpretability. This study leverages eXtreme Gradient Boosting (XGBoost) to detect blowing snow events using meteorological and snow flux monitoring data from three weather stations in the Alps. Through 5-fold cross-validation, the model achieved impressive performance metrics, with precision rates exceeding 0.94 for non-blowing snow events and 0.77-0.80 for blowing snow events. The SHAP framework was employed to analyze the relative importance of meteorological factors, revealing that maximum wind speed (WS-MAX), average wind speed (WS-AVG), air temperature (AT), and relative humidity (AH) are the most influential factors. Additionally, Partial dependence plots (PDP) demonstrated a linear correlation between increased WS-MAX and the probability of blowing snow, while WS-AVG showed diminishing returns beyond 10 m/s. Notably, AT below -3°C strongly correlates with blowing snow occurrence, whereas AT above -3°C exhibits a negative relationship. Relative humidity plays a significant role, with values exceeding 60% stabilizing the probability of blowing snow, peaking near 100%. This research contributes to drifting snow event dynamics by integrating explainable artificial intelligence techniques (XAI), thereby improving model interpretability and supporting data-driven decision-making in meteorological applications.

## 1 Introduction

### 1.1 Motivation

Wind-blown snow is a natural hazard caused by strong winds and a large amount of drifting snow particles, particularly common in cold, snowy open areas [1]. When the surface snow particles are acted upon by the wind, they are lifted into the air, where they collide and adhere to one another. When the energy of the wind-blown snow flow is relatively low, these airborne snow particles descend along a parabolic trajectory, eventually landing back on the ground, creating a cycle composed of initiation, drifting, and reattachment processes

**Data availability statement:** The data underlying the results presented in the study are available from the ISAW online database at http://iav-portal.com/index.php?nav=iodmisawlist. Additionally, the cleaned dataset in S Files 1, 2, and 3 and all author-generated code in S4 File are provided in the Supporting Information to support reproducibility and further research.

**Funding:** The author(s) received no specific funding for this work.

**Competing interests:** NO authors have competing interests.

[2]. Wind-transported snow can redistribute snow accumulation, leading to the formation of uneven snowpacks, with resulting snow depths typically ranging from 3 to 8 times greater than natural snow depth [3].

The excessive snow accumulation caused by windblown snow can have serious impacts on human activities, potentially triggering secondary disasters such as avalanches [4,5]. Along high-altitude mountain roads, windblown snow can significantly reduce visibility, creating challenges for drivers [6,7], while also leading to the formation of secondary snow on road surfaces, severely diminishing transportation capacity and, in extreme cases, causing traffic blockages [8]. The occurrence of windblown snow largely depends on a combination of environmental meteorological conditions (such as wind speed and temperature), snow properties, and vegetation characteristics [9]. The influence of these factors complicates the snow drifting process, making it essential to accurately quantify their effects. This is crucial for accurately forecasting windblown snow events and reducing the potential for loss of life, property damage, and transportation disruption.

## 1.2 Review of literature

Existing research has found that in cold, snowy regions, wind is one of the most important driving forces behind snow accumulation patterns [10]. The wind speed at which blowing snow begins is referred to as the threshold wind speed, a critical parameter for predicting the initial movement of snow particles. In most current models for assessing windblown snow events, the module for calculating snow transport is activated only when wind speed exceeds a threshold value [11]. Moreover, the neglect or simplification of the effects of wind redistribution of snow has been identified as a major source of uncertainty in simulations of snow cover evolution [12]. Therefore, accurately estimating the threshold wind speed for snow transport is key to predicting the occurrence of windblown snow and quantifying its subsequent impacts.

Research on the conditions for windblown snow occurrence, specifically the threshold wind speed, has utilized various specialized instruments to directly measure blowing snow within localized areas. For instance, Budd et al. employed mechanical collectors [13], while optical sensors (snow particle counters, SPC) have been deployed in Antarctica and the Alps [14,15]. Additionally, acoustic sensors like FlowCapt have been used to provide relatively reliable measurements of blowing snow mass flux [16].

Early research made significant strides in determining the threshold wind speed for snow transport. For example, Berg found that in Niwot Ridge, Colorado, wind speeds at a height of 1 meter need to reach 4-6 m/s to initiate snow transport [17]. As research progressed, it became evident that when snow becomes wet, the adhesive forces significantly increase, as meltwater enhances the cohesion between particles [18]. The sintering process of snow particles also plays a substantial role in the development of adhesive strength [19]. Li & Pomeroy (1997) identified a threshold wind speed range for wet snow transport of 7 to 14 m/s, noting that the threshold for dry snow transport is over 2 m/s lower than that for wet snow [18]. Additionally, Liston et al. determined that the typical threshold wind speed for recently fallen or slightly aged cold dry snow is between 4 and 5 m/s [20]. Although early studies generally assumed a constant threshold wind speed, significant variations exist among different studies, primarily attributed to factors such as wind speed, temperature, and snow characteristics [6].

To better parameterize the occurrence of snow drift, subsequent research has explored the relationship between threshold wind speed and the microstructural characteristics of surface snow, proposing alternative methods using empirical formulas. Some studies have attempted to correlate threshold wind speed with snow particle shape deformation [21], snowpack

density [22], and the diameter of snow clumps formed by the aggregation of snow particles [23]. These efforts have led to the establishment of numerous empirical formulas that describe the relationship between threshold wind speed and snow layer conditions. Such parameterizations are widely applied in numerical models to characterize wind-driven snow transport processes, thereby providing a foundation for the development of snow drift models [24,25].

Research on the relationship between threshold wind speed and meteorological conditions faces additional challenges. Firstly, the occurrence of snow drift is closely linked to local microclimate conditions (temperature, humidity, wind speed, etc.), which are unevenly distributed in space and exhibit complex interactions, making it difficult to establish universally applicable models [26]. Furthermore, variations in topography, vegetation cover, and other environmental features across different regions also influence the formation and development of snow drift, thereby increasing research complexity. As a result, many parameterization studies related to meteorological conditions have only been validated in limited areas [27], restricting their applicability. Additionally, a significant challenge researchers face is the lack of a standardized method to determine the threshold wind speed at which snow drift events occur [28]. Field measurements may indicate that the moment of non-zero snow flux corresponds to the occurrence of a snow drift event; however, determining the corresponding threshold wind speed at that moment is ambiguous. It remains unclear whether the relevant wind speed should be the one at that moment (which is greater than the threshold), the preceding moment (which is less than the threshold), or a weighted sum of both. This dilemma complicates the accurate measurement of threshold wind speed. The absence of standardization not only leads to discrepancies in threshold wind speed estimates but also hinders the applicability of these findings to broader regions or climatic conditions.

To address the issues inherent in traditional research methods, the introduction of machine learning techniques offers a novel approach. Combining XGBoost with SHAP analysis allows for flexible adjustments across different regions and temporal scales, adapting to variable meteorological conditions. The advantage of this approach lies in its departure from reliance on single empirical formulas; instead, it is grounded in actual observational data, thereby enhancing the model's applicability and accuracy.

XGBoost (eXtreme Gradient Boosting), initially proposed by Chen and Guestrin in 2016, is a relatively new method that demonstrates high accuracy, fast processing times, and lower computational costs and complexity [29]. The model's strong performance has been validated in predicting the occurrence and duration of various events using multiple data sources [30]. Moreover, XGBoost outperforms several other machine learning techniques in predicting the likelihood of target events, including logistic regression, Bayesian regularized neural networks, Pegasos SVM, Bagging average neural networks, deep neural networks, and traditional Gradient Boosting methods [31].

Meanwhile, machine learning models often have limitations, particularly in their interpretability; their "black box" nature makes it challenging for users to understand how specific predictions are derived. To address this issue, recent studies have begun utilizing SHAP values to assess feature importance within models. SHAP values provide a systematic approach to quantify the contribution of each feature to the final prediction. Lundberg and Lee developed a Python package capable of calculating SHAP values for various machine-learning models [32], offering more nuanced analyses for feature evaluation in susceptibility studies related to snow avalanches [33], wildfires [34], and flooding [35].

## 1.3  Contributions

Despite progress in research on forecasting snow drift events, several challenges remain in establishing the relationship with meteorological conditions. One major issue is that existing studies often rely on traditional empirical models that oversimplify the complex interactions

between meteorological variables, leading to significant discrepancies in threshold wind speed estimates across different regions and climatic conditions. These traditional approaches typically treat wind speed and other environmental factors as isolated variables, overlooking their dynamic and multifaceted interactions. As a result, threshold wind speed values, which are critical for predicting snow transport and drift initiation, often vary widely between studies.

Moreover, existing research generally lacks a standardized method for defining and measuring threshold wind speeds. Field-based studies have reported widely differing values for the wind speed required to initiate snow transport, with discrepancies often attributed to differences in experimental setup, measurement techniques, and regional environmental conditions. This inconsistency not only complicates the validation of snow drift models across different regions but also limits their applicability to broader climatic contexts. As a result, current models often fail to generalize effectively, making it difficult to predict snow drift behavior reliably in diverse environments.

These challenges underscore the need for a more nuanced, data-driven approach that can better account for the complex, multifactorial nature of snow drift dynamics. To tackle these issues, the main objective of this study is to assess the performance of XGBoost in detecting the occurrence of blowing snow and to analyze the importance of individual features for detection using SHAP. The training data consists of over a decade of continuous multi-factor field meteorological conditions and snow drift mass flux measurements from three ISAW stations in the Alpine region. To address the issue of data imbalance, the Synthetic Minority Oversampling Technique (SMOTE) was employed to prepare the dataset, thereby enhancing the model's robustness. By modelling based on actual observational data, the study also utilizes Partial Dependence Plots (PDP) to analyze the dependencies between features, providing a clear visualization of how meteorological characteristics influence threshold wind speed. The knowledge gained from this study can offer a more scientific basis for predicting and assessing snow drift risks and provide new perspectives and methods for future research.

The rest of this article is organized as follows: Section 2 presents the data sources utilized in this study and details the data preprocessing procedures. Section 3 provides an in-depth review of XGBoost and SHAP. In Section 4, the performance of the blowing snow detection model is analyzed and discussed, along with a comprehensive interpretation of features using SHAP and PDP. Finally, Section 5 outlines the findings, limitations, and future directions of the study.

## 2 Data and variables

### 2.1 Training data sources

The training data used in this study was derived from field observations in the European Alps. Since 2004, IAV Engineering Ltd. has established meteorological stations in the region to conduct long-term observations of drifting snow. The data has been publicly available through their online database, ISAW (http://iav-portal.com/index.php?nav=iodmisawlist). As of December 2023, 14 out of the 21 meteorological stations set up by IAV in this region continue to provide real-time data. The long-term, stable, and accessible nature of ISAW data has made it a valuable resource for windblown snow research, particularly for studies focused on the conditions under which windblown snow events occur [19].

Meteorological data and snow flux were recorded at each station with an hourly resolution. The specific variable names and their descriptions are listed in Table 1. This study focused on a more selective set of parameters, including wind speed, temperature, humidity, precipitation, etc. While several additional variables, such as the angle of repose, snow particle size distribution, snow density, and thermal permittivity, have been identified in the literature

**Table 1. Variables and definitions of field monitoring data.**

| Observed variables | Unit | Definition |
|---|---|---|
| WS-AVG | m/s | The average wind speed within an hour |
| WS-MAX | m/s | The highest wind speed value within the hour |
| WD-MIN | ° | The minimum wind direction value within the hour |
| WD-MAX | ° | The maximum wind direction value within the hour |
| RYS | mm | The cumulative precipitation from the beginning of the year to the current time (Cleared on May 1) |
| RAS | mm | The total precipitation within the observation period (the hour) |
| RAI-6 | mm/h | The total precipitation intensity during the observation period and the preceding 5 hours |
| RAI-24 | mm/h | The total precipitation intensity during the observation period and the preceding 23 hours |
| RI-MIN | mm/h | The minimum precipitation intensity recorded within the hour |
| RI-MAX | mm/h | The maximum precipitation intensity recorded within the hour |
| SH-SEG1 | mm | The snow height in the SEG1 area |
| SH-SEG2 | mm | The snow height in the SEG2 area |
| AT | °C | The average air temperature within the hour |
| AH | % | The average relative humidity within the hour |
| SD | g/m²·s | The average snow flux within the hour |

as influencing snow characteristics [36], they were not considered in the present study. This choice was primarily driven by the practical considerations of real-time field application, as many of these additional variables require specialized instruments or complex measurement techniques, which could limit the model's operational applicability. Moreover, many of the variables relevant to snow drift prediction are interrelated, which introduces further complexity in their integration and interpretation within a predictive framework.

In Table 1, wind speed and direction were measured at a height of 3.5 meters using the Wind Monitor (YOUNG Model 05103). Precipitation intensity was recorded using a Campbell Scientific ARG100 tipping-bucket rain gauge, while snow depth was measured with an SR50 ultrasonic sensor. Snow depth measurements were taken from two different areas near the meteorological stations to account for the spatial variability of snow accumulation and to improve the representativeness of the results, which can be less reliable when measured at a single point.

The critical parameter for characterizing windblown snow, snow flux, is measured using the FlowCapt. This acoustic sensor is designed to continuously and automatically record airborne snow flux by detecting sound generated by snow particles colliding with the instrument's tube [37]. The sensor consists of a tube approximately 1 meter in length, containing electroacoustic transducers that omnidirectionally capture the acoustic vibrations caused by wind-transported snow particles hitting the tube. Through electrical filtering and spectral analysis, the device effectively and accurately differentiates between the low-frequency noise generated by wind and the high-frequency signal from windblown snow [38].

Although some studies have raised concerns about the accuracy of FlowCapt sensors in quantitatively measuring snow flux [39,40], research by Trouvilliez et al. suggests that, despite its limitations in precise snow flux measurements, the FlowCapt sensor shows excellent potential for long-term, continuous monitoring of drifting and blowing snow indices [16]. As one of the few instruments capable of operating across a wide range of climatic conditions, the FlowCapt sensor can continuously and automatically monitor snow drift as long as the sensor remains partially exposed.

It is important to note that not all meteorological stations possess the variables listed in Table 1. Only five stations—FBER, FCMB, FGIE, FHUE, and FMOR—include all the parameters shown in the table. Since this study focuses on the impact of these features, the final selection consists of the three stations among the five total meteorological stations with the most extended data collection periods (FGIE, FBER, FHUE). Their locations are illustrated in Fig 1, situated in the mountainous region between Lyon, France, and Torino, Italy. Detailed information is provided in Table 2, where the average elevation is approximately 2000 meters, and the period of the station data covers 13 years (2011-2023).

## 2.2 Data analysis and cleaning rules

To facilitate the establishment of data cleaning rules and gain a more detailed understanding of the parameter variations within the dataset, Fig 2 presents the changes in several parameters recorded at the FGIE station over approximately 10 days in mid-March 2019. The parameters include wind speed, snow flux, snow depth, precipitation, air temperature, and humidity.

An initial analysis of the snow flux (SD) data revealed four significant wind-blown snow events during this period. Additionally, two notable snowfall occurrences were observed during the third and fourth wind-blown snow events, each with hourly precipitation (RAS) exceeding 5 mm. During the wind-blown snow events, the air temperature (AT) remained below -5°C, and the humidity (AH) was consistently at or near the maximum value of 100%.

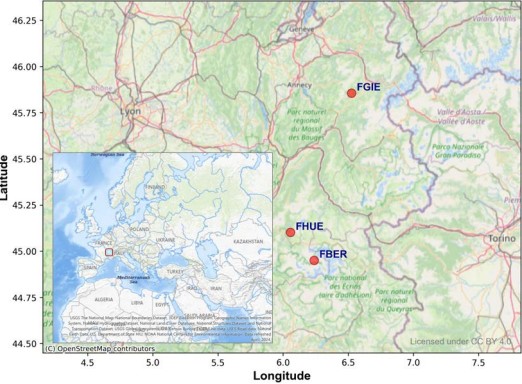

**Fig 1. Study area and locations of selected meteorological stations.**

**Table 2. List of stations used in this study.**

| | Station information | | |
|---|---|---|---|
| Site name | FBER | FGIE | FHUE |
| Country | France | France | France |
| Region | Isère | Savoie | Isère |
| Altitude (m) | 2390 | 1812 | 2064 |
| Latitude (°N) | 44° 56' 59.8" | 45° 51' 19.0" | 45° 6' 6.7" |
| Longitude (°E) | 6° 14' 13.5" | 6° 31' 32.5" | 6° 3' 22.2" |
| Data period | From Jan. 2011 to Dec. 2023. | | |

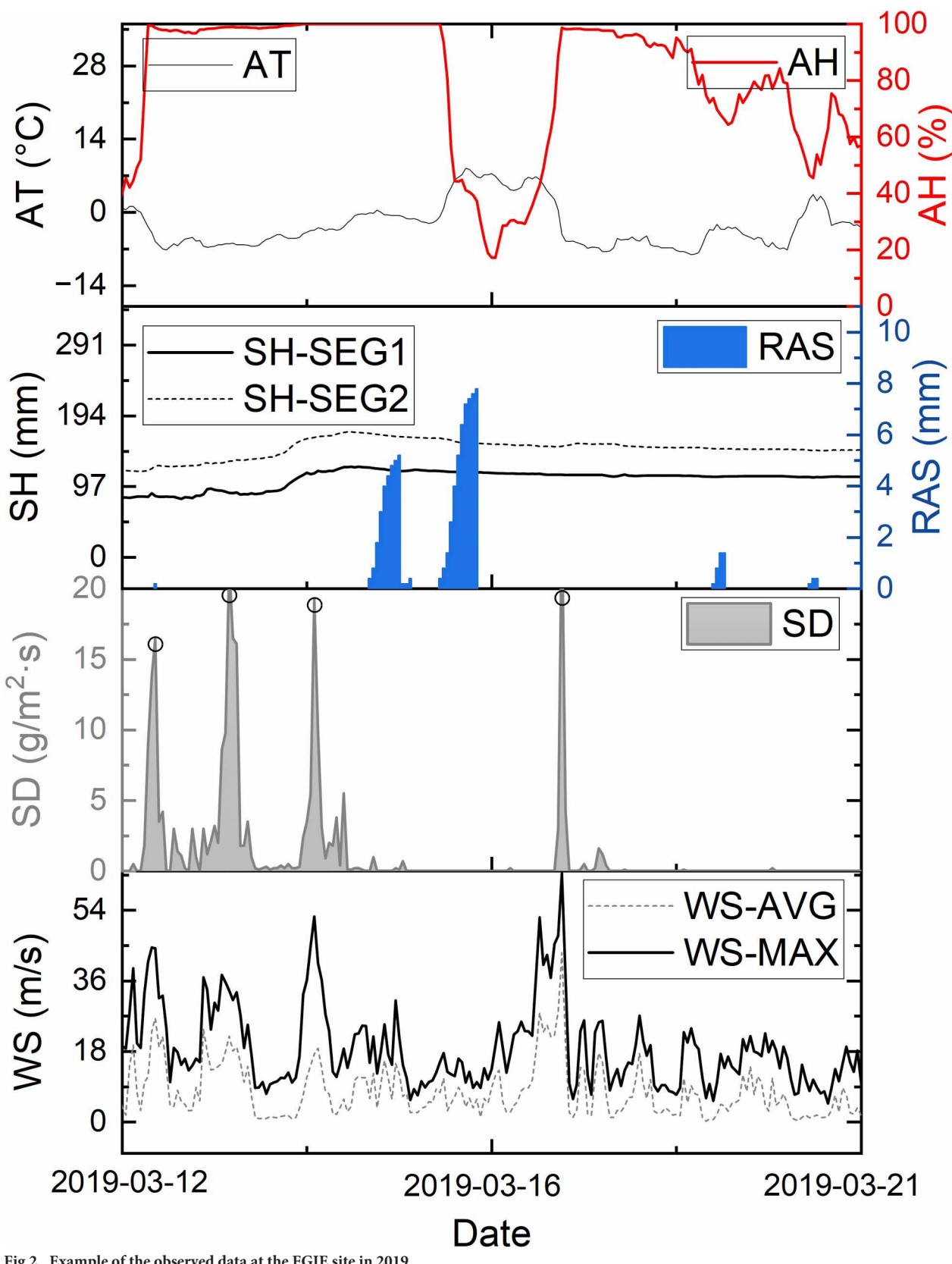

**Fig 2. Example of the observed data at the FGIE site in 2019.**

During the wind-blown snow events, both the maximum and average wind speeds (WS) exhibited peak values, with the maximum wind speed exceeding 20 m/s. The snow depth in the two observation areas differed by approximately 50 cm, with average snow depths of 100 cm and 150 cm, respectively. However, the snow depth (SH) in both areas followed a similar trend: after the first three wind-blown snow events, a significant increase in snow depth was observed, followed by a gradual decrease due to melting.

Notably, snowfall events did not significantly increase snow depth in the observation areas compared to the wind-blown snow events. Although these results are based on a partial dataset, this pattern was observed multiple times throughout the study. This suggests that, to a considerable extent, the changes in snow depth in this region are primarily driven by wind-blown snow rather than by snowfall events.

Upon further examination of the raw meteorological station data, missing values, outliers, or inconsistencies were common in the recorded measurements. These issues can significantly impact the accuracy and predictive power of models. Therefore, data cleaning is a crucial step in ensuring reliable results. This process involves removing or correcting outliers that violate physical laws, eliminating noisy data, and preserving key information to enhance the model's predictive performance and produce more interpretable results.

The first step in data cleaning was to remove outliers in wind speed and air temperature/humidity. Specifically, we discarded samples where wind speed exceeded 40 m/s, humidity was less than 0%, or temperature exceeded 40°C or dropped below -40°C.

Due to snow accumulation and melting, changes in the exposed length of the FlowCapt sensor tube can introduce uncertainty in the measurement of snow mass flux. As a result, the observed snow mass flux is more suitable for qualitative assessments of wind-blown snow events rather than for quantitative intensity analysis. In this study, samples with a snow flux greater than $0.1\,g/m^2{\cdot}s$ were defined as wind-blown snow events [19,41], with a wind-blown snow index assigned a value of 1. All other samples were classified as non-blown snow events and assigned a value of 0.

Previous studies have shown that the FlowCapt sensor is sensitive to moving soil particles, which can lead to false detections of wind-blown snow events, introducing significant uncertainty into the monitoring data [5]. By analyzing snow depth data from different years at various stations, it was found that snow depth remained nearly zero between May and October each year. Therefore, only data from the winter and spring seasons (November to April) were used in this study, minimizing the uncertainty caused by false detections related to soil particles.

Detailed supplementary information is essential to enhance transparency and provide a thorough understanding of the dataset, as well as its relevance to the development of the wind-blown snow prediction model. Specifically, Supporting Information S1 Table summarizes the statistical characteristics of the cleaned data, prepared in accordance with the defined processing criteria. Furthermore, S2 Table presents the Spearman correlation coefficients between the analyzed parameters and the outcomes of snow-blowing events, providing insights into the relationships influencing the observed phenomena.

Additionally, resampling was also applied to the cleaned dataset obtained using the cleaning rules. The Synthetic Minority Over-sampling Technique (SMOTE) was employed to address the issue of wind-blown snow events being underrepresented in the overall dataset. Since wind-blown snow events belong to the minority class, models trained on unbalanced data tend to favor predicting the majority class, which leads to better performance in predicting non-blown snow events but poorer performance in detecting wind-blown snow events. By using SMOTE to generate additional minority class samples, the dataset was balanced, improving the model's sensitivity to wind-blown snow events and enhancing its overall

classification performance. After resampling the cleaned dataset, an 80-20 split was applied to create training and validation sets. Specifically, 80% of the resampled data was allocated to the training set, and the remaining 20% was reserved for the validation set.

## 3 Methodology

In this study, XGBoost was employed to model the prediction of wind-blown snow events. XGBoost is an efficient machine learning algorithm based on Gradient Boosting Decision Trees (GBDT) and is widely used in classification tasks due to its outstanding performance and training speed. Decision Trees (DT) are tree-structured models that typically use simple decision rules, starting from the root node and proceeding through a series of internal nodes to reach the leaf nodes. GBDT is an ensemble learning method consisting of a sequence of DTs, where each tree learns from the previous one and influences the construction of subsequent trees, gradually improving the model to build a more powerful learner [42].

Data processing and modeling were conducted in a Python 3.12 environment using the Scikit-learn library for model training and evaluation. Additionally, to enhance the interpretability of the model, SHAP (SHapley Additive exPlanations) values were used to gain deeper insights into the importance and impact of each feature in the model's predictions.

### 3.1 XGBoost: eXtreme gradient boosting

XGBoost, introduced by Chen and Guestrin [29], stands for "eXtreme Gradient Boosting." It is an extended and optimized version of Boosting Trees. Compared to traditional Gradient Boosting Decision Trees (GBDT), XGBoost incorporates several improvements in optimization efficiency, model generalization, and training speed, achieving the engineering goal of fast computation and excellent performance.

The fundamental building blocks of XGBoost are decision trees, which serve as "weak learners." These trees are constructed sequentially, with each subsequent tree considering the prediction results of the previous tree. Specifically, the bias from the prior tree is taken into account, giving more attention to the training samples that were incorrectly predicted. This process adjusts the sample distribution before training the next tree. XGBoost uses K additive trees to approximate the predicted output $\hat{y}_i$ as follows:

$$\hat{y}_i = \sum_{k=1}^{K} f_k(X_i), f_k \in F \tag{1}$$

Here, $f_k$ is an independent classification and Regression Tree (CART) at each of $k$ steps which maps the input variables $X_i$ to $y_i$, and $F$ is the space of functions containing all CARTs.

To optimize the model structure, XGBoost introduces a regularization term that penalizes model complexity in the objective function, thereby reducing the risk of overfitting. The optimized objective function can be expressed as:

$$L(\theta) = \sum_{i=1}^{n} l\left(y_i, \hat{y}_i^{(t)}\right) + \sum_{t}^{k=1} \Omega(f_k) \tag{2}$$

where $l$ represents the loss function, indicating the error between the model's predicted values and the true values, and $\Omega(f_k)$ is the regularization term, defined as follows:

$$\Omega(f_k) = \gamma T + \frac{1}{2} \lambda \sum_{j=1}^{T} w_j^2 \tag{3}$$

where $T$ represents the number of leaf nodes in the tree, $w_j$ represents the weight of the j-th leaf node. $\lambda$ is the penalty term for the weight of the leaf nodes, and $\gamma$ controls the complexity of the tree.

By incorporating the regularization term, XGBoost effectively avoids the issue of overfitting, whereas traditional GBDT lacks this component and relies solely on hyperparameter tuning to control model complexity.

Another significant improvement of XGBoost is its use of both the first and second-order derivatives of the loss function to optimize the objective function. By performing a second-order Taylor expansion on each new tree, the model can more accurately update the weights during each iteration, thereby improving training efficiency. The optimized objective function can be expressed as:

$$L^{(t)} = \sum_{i=1}^{n}\left[g_i f_t\left(x_i\right) + \frac{1}{2}h_i f_t^2\left(x_i\right)\right] + \Omega\left(f_t\right) \tag{4}$$

where $g_i$ represents the first-order derivative of the loss function, and $h_i$ represents the second-order derivative.

During the tree construction process, XGBoost automatically handles missing values by determining the optimal split direction for samples with missing data. At each split point, the model calculates the gain when assigning missing samples to either the left or right node, selecting the direction that yields the highest gain. In addition to handling missing values, XGBoost optimizes tree structures by employing a post-pruning mechanism. After constructing the tree to its maximum depth, it decides whether to prune the tree based on the node's gain value. If the gain is below a certain threshold, the node is pruned, effectively controlling the complexity of the tree. In contrast, Gradient Boosting Decision Trees (GBDT) use pre-pruning, which restricts tree flexibility by limiting the maximum depth of the tree in advance.

To further improve training efficiency, XGBoost leverages a column block approach for parallel computing. Specifically, the model stores features by columns and calculates gradients and Hessians independently on column blocks, allowing for parallel processing of multiple features in a multithreaded environment. This enhances computational efficiency and scalability.

With these improvements, the XGBoost algorithm can efficiently handle ISAW's large-scale field monitoring data, which includes multiple complex and interrelated meteorological features [43]. The second-order derivative optimization mechanism allows XGBoost to capture the nonlinear relationships between features more accurately. Additionally, XGBoost's built-in missing value handling mechanism performs excellently when dealing with missing records in meteorological data. Without requiring manual imputation or data removal, the model automatically selects the optimal split direction, maximizing data utilization. This makes XGBoost highly suitable for such complex datasets.

During the training of the XGBoost model, adjusting multiple parameters is crucial for maximizing model performance, particularly to prevent overfitting and model complexity. To further optimize performance, this study employed GridSearchCV for grid search combined with 5-fold cross-validation. This technique divides the dataset into five equal subsets, where each subset is used as a validation set while the remaining four are used for training in an iterative manner. By cycling through all subsets, the method ensures that every data point is tested exactly once, providing a comprehensive evaluation of model performance [44]. This approach ensures robust evaluation by mitigating overfitting and providing reliable performance estimates on unseen data. The grid search specifically targeted optimizing the following parameters: n_estimators (number of trees), max_depth (depth of trees), and learning_rate.

Through the grid search process, the model automatically explored the pre-defined parameter space to determine the optimal values. Based on these optimal parameters, the final XGBoost model was trained. After resampling the data, the model achieved an optimal level of log loss, reflecting its improved performance in handling complex datasets.

### 3.2 Model interpretation

Efficient classification relies on robust feature extraction techniques to ensure accurate and reliable model performance. In this paper, the Shapley additive Interpretation (SHAP) proposed by Lundberg and Lee is adopted as a tool for feature importance analysis [32]. SHAP is based on game theory [45] and local explanations [46], and it offers a means to estimate the contribution of each feature. Assume an XGBoost model where a group N (with p features) is used to predict an output; Shapley values are determined through:

$$Shapley\left(X_j\right) = \sum_{S\subseteq N\backslash(j)} \frac{|S|!\left(p-|S|-1\right)!}{p!}\left[f\left(S\cup\{j\}\right)-f\left(S\right)\right] \tag{5}$$

where Shapely($X_j$) is contribution of feature $X_j$, N\{j} is a set of all possible combinations of features excluding $X_j$, S is a feature set in N\{j}, f(S) is the model prediction with features in S, and $f(S\cup\{j\})$ is the model prediction with features in S plus feature $X_j$.

SHAP values are particularly useful for evaluating decision-tree-based models. They provide clear insights into how each feature contributes to a model's final prediction, offering a transparent way to understand the influence of individual variables. As shown in Fig 3, for each prediction of the probability of drifting snow occurrence, SHAP values clearly illustrate how factors such as wind speed and temperature specifically influence the final predicted probability.

The base rate (0.1) represents the model's output when no features are influencing the prediction. As different features exert varying degrees of influence on the prediction, the SHAP values provide insight into how the cumulative effect of these variables leads to the final output, which, in this case, is the probability of drifting snow. SHAP values thus clarify which features drive the model's decisions (Output=0.7), making the model more interpretable and aiding in identifying key predictors.

After ranking the importance of different features using the calculated SHAP values, this study introduced Partial Dependence Plots (PDP) to analyze further how the top features

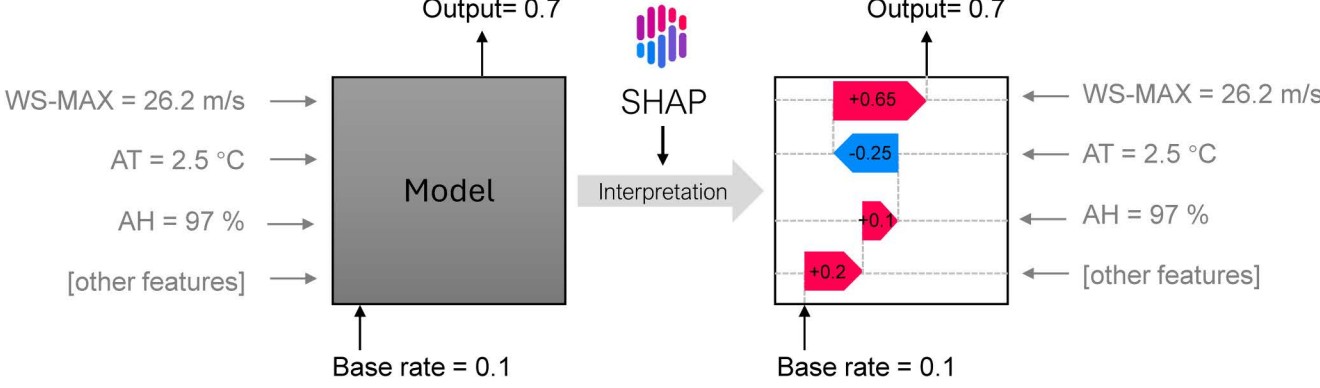

**Fig 3. The SHAP interpretation for the model in a single prediction example.**

individually and collectively influence the model's predictions. PDP examines the impact of changes in a target feature on the model output while keeping other features constant, thereby revealing how the model makes predictions based on specific features. This method is particularly useful for interpreting nonlinear and complex models, such as XGBoost, by providing a simplified understanding of the model's dependence on specific features. The Partial Dependence Function (PDF) is defined as follows:

$$\hat{f}_j\left(x_j\right)=\mathbb{E}_{x_{-j}}\left[\hat{f}\left(x_j,x_{-j}\right)\right] \tag{6}$$

where $\hat{f}\left(x_j,x_{-j}\right)$ represents the model's prediction function, and $\mathbb{E}_{x_{-j}}$ denotes the expectation over the marginal distribution of all other features $x_j$, except for $x_j$. In this form, the PD plot shows the marginal effect of feature $x_j$ on the predicted outcome while averaging out the effects of the other features.

## 4  Result

### 4.1  Model evaluation

This study selected three commonly used evaluation metrics (precision, recall, and F1-score) to assess the model's predictive performance. The calculation of these metrics is based on the confusion matrix, as defined in Table 3. After a classification task is completed, the samples are divided into four categories: True Positives (TP), False Positives (FP), True Negatives (TN), and False Negatives (FN).

The relevant definition of Precision, Recall, and F1 score is as follows.

$$\begin{cases} Precision=\dfrac{TP}{TP+FP} \\ Recall=\dfrac{TP}{TP+FN} \\ F1-score=2\times\dfrac{Recall*Precision}{Recall+Precision} \end{cases} \tag{7}$$

Based on the detection data from three meteorological stations selected from ISAW, XGBoost was trained on 80% of randomly selected data, with the remaining 20% used to test the model. Additionally, to evaluate the stability of the model's performance, a 5-fold cross-validation was conducted. The training data was randomly divided into five subsets, with four subsets used for training and one subset used for testing. Table 4 presents the results of the three key evaluation metrics—Precision, Recall, and F1-score—after 5-fold cross-validation. In these results, Class 0 represents non-wind-blown snow events, while Class 1 represents wind-blown snow events.

FBER-o refers to the original dataset, which was not subjected to resampling prior to training, while the other datasets were resampled using SMOTE. FBER-a and FBER-t represent datasets processed with ADASYN and Tomek Links, respectively. Both ADASYN and SMOTE

**Table 3.  Confusion matrix of a binary classification model.**

| True Condition | Predicted Condition | |
|---|---|---|
| | **Positive** | **Negative** |
| Positive | True Positive (TP) | False Negative (FN) |
| Negative | False Positive (FP) | True Negative (TN) |

**Table 4. Performance of the models on field monitoring data.**

| | | Class 0 | | | Class 1 | | |
|---|---|---|---|---|---|---|---|
| | Dataset | Precision | Recall | F1-score | Precision | Recall | F1-score |
| XGBoost | FBER | 0.97 | 0.95 | 0.96 | 0.79 | 0.86 | 0.82 |
| | FGIE | 0.94 | 0.96 | 0.95 | 0.77 | 0.84 | 0.80 |
| | FHUE | 0.98 | 0.96 | 0.97 | 0.80 | 0.87 | 0.83 |
| | FBER-a | 0.98 | 0.96 | 0.97 | 0.77 | 0.88 | 0.82 |
| | FBER-t | 0.97 | 0.98 | 0.98 | 0.85 | 0.82 | 0.83 |
| | FBER-o | 0.95 | 0.98 | 0.96 | 0.83 | 0.63 | 0.72 |
| SVM | FBER | 0.95 | 0.99 | 0.97 | 0.77 | 0.45 | 0.57 |
| RF | FBER | 0.95 | 0.98 | 0.96 | 0.68 | 0.79 | 0.73 |
| GBDT | FBER | 0.98 | 0.97 | 0.97 | 0.71 | 0.81 | 0.76 |

fall under the category of oversampling techniques, with ADASYN being a more advanced approach. In contrast, Tomek Links is an undersampling method. All three are widely used techniques for addressing imbalanced datasets [47]. Unless otherwise specified, datasets are assumed to have been processed using SMOTE. Furthermore, data from the FBER station was selected to compare the performance of the model without SMOTE sampling against other deep learning models on this dataset, such as SVM, RF, and GBDT.

A comprehensive analysis of the data in Table 4 reveals that almost all models perform exceptionally well in predicting Class 0, with Precision and Recall both exceeding 0.94, and F1-scores ranging between 0.92 and 0.97. This indicates that the models are highly accurate in predicting non-wind-blown snow events, and most models exhibit stable performance on these samples. Notably, XGBoost and GBDT perform the best, with both Precision and Recall approaching 1.

Due to the class imbalance in wind-blown snow events (Class 1), the models face greater difficulty in predicting these events, especially with significant discrepancies between Precision and Recall. Among the models, XGBoost demonstrates the most stable performance when handling Class 1. Across the three resampled datasets (FBER, FGIE, FHUE), Precision ranges from approximately 0.77 to 0.80, Recall from 0.84 to 0.87, and the F1-score is around 0.80 to 0.83.

To provide a more intuitive comparison of the performance of different models on the same dataset, supplementary figures (S1 and S2 Figs) have been included in the Supporting Information section. S1 Fig. presents the wind-blown snow events (gray) and non-wind-blown snow events (white) within the validation set, which accounts for 20% of the total dataset (FBER). Each unit cell in the figure represents one hour. S2 Fig. compares the predicted results from XGBoost, SVM, RF, and GBDT models against the observed validation set. In this figure, white cells represent instances where the predicted data matches the observed data, while red cells indicate mismatches. The results clearly show that XGBoost demonstrates superior predictive performance compared to the other models, as evidenced by its higher agreement with the observed data.

The superior performance of XGBoost can be attributed to several key factors, especially when compared to models such as SVM, RF, and GBDT. XGBoost utilizes an ensemble learning framework, which aggregates multiple weak learners (decision trees) to enhance predictive accuracy. Furthermore, XGBoost incorporates an iterative correction process in which each tree not only focuses on correcting the errors from the current dataset but also refines the mistakes made by previous trees. This mechanism proves particularly effective for handling

imbalanced datasets, such as in the case of wind-blown snow events, where the target class (Class 1) is underrepresented. By progressively refining the model's predictions, XGBoost is better equipped to capture complex patterns and subtle feature differences, especially in smaller sample sizes. This iterative adjustment enables the model to handle the challenges posed by class imbalance and ensures that it can identify minority class events with greater accuracy.

The performance of XGBoost on the original FBER-o dataset for Class 1 predictions differs significantly from that on the resampled datasets. After resampling, although the Precision decreased slightly from 0.83 to 0.79–0.80, the Recall improved markedly, rising from 0.63 to 0.84–0.87. This indicates that resampling allows the model to better capture the minority class of wind-blown snow events and significantly reduce the number of False Negatives (missed wind-blown snow events). This aligns with the characteristics of wind-blown snow hazard prediction, as these events tend to occur infrequently but can cause severe damage. In such cases, the consequences of missing a wind-blown snow event (False Negative) could be more severe than a false alarm (False Positive).

Furthermore, a comparison of different resampling methods revealed that ADASYN slightly outperformed SMOTE in terms of Recall, while Tomek Links achieved marginally higher Precision. Despite these differences, the average performance, as measured by the F1-score, was comparable across the three methods. Notably, all three techniques resulted in approximately a 15% improvement in F1-score compared to the original dataset (FBER-o).

As shown in Fig 4, the yearly variation of the model's performance metrics is illustrated, based on data from the FBER station. The metrics, including precision, recall, and F1-score, are presented for both Class 1 (a) and Class 0 (b). In these figures, the solid lines represent the performance metrics for models trained on individual yearly datasets (Per-Year Data), while the dashed lines correspond to the metrics obtained from the 13-year aggregated dataset (13-Year Data). Additionally, the red bars indicate the number of snowdrift samples (Class 1) available in each year.

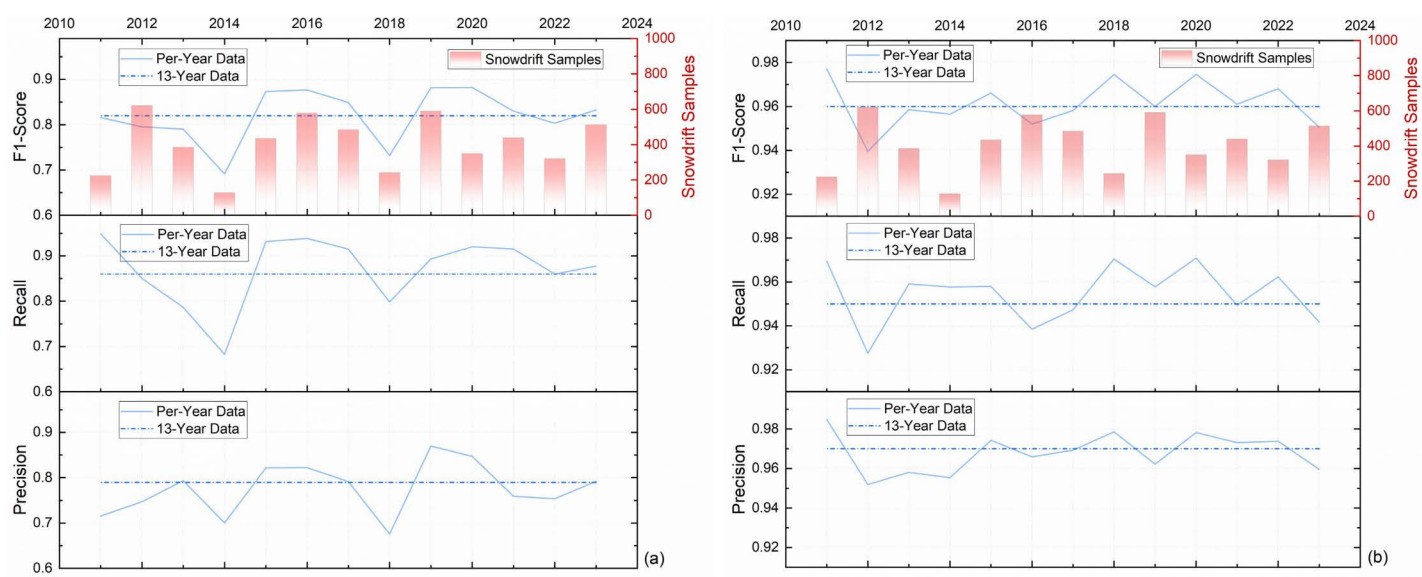

**Fig 4. Yearly variation of model performance metrics based on FBER station data: (a) Class 1, (b) Class 0.**

From [Fig 4a], it can be observed that the model's performance for Class 1 exhibits fluctuations in F1-score and recall across different years, which are closely related to the yearly variations in snowdrift sample size. For example, peaks in sample size, such as those in 2012 and 2017, correlate with higher F1-scores, whereas years with lower sample sizes, such as 2020, show a decline in these metrics. In contrast, [Fig 4b] shows that the model's performance for Class 0 remains relatively stable across all years, with minimal variations in precision, recall, and F1-score. The consistency of these metrics indicates that the model's predictions for non-blowing snow events are less sensitive to temporal dependencies and sample size variability. Furthermore, as the sample size has a significant impact on the results, the 13-year dataset was selected as the training data, rather than using data from a single year or a subset of the years.

## 4.2 Feature analysis

To explain the optimal performance of the XGBoost model on the datasets, SHAP was used to quantify the contribution of each feature to the model's output for individual instances. The global feature importance was ranked based on the average absolute SHAP values (Mean

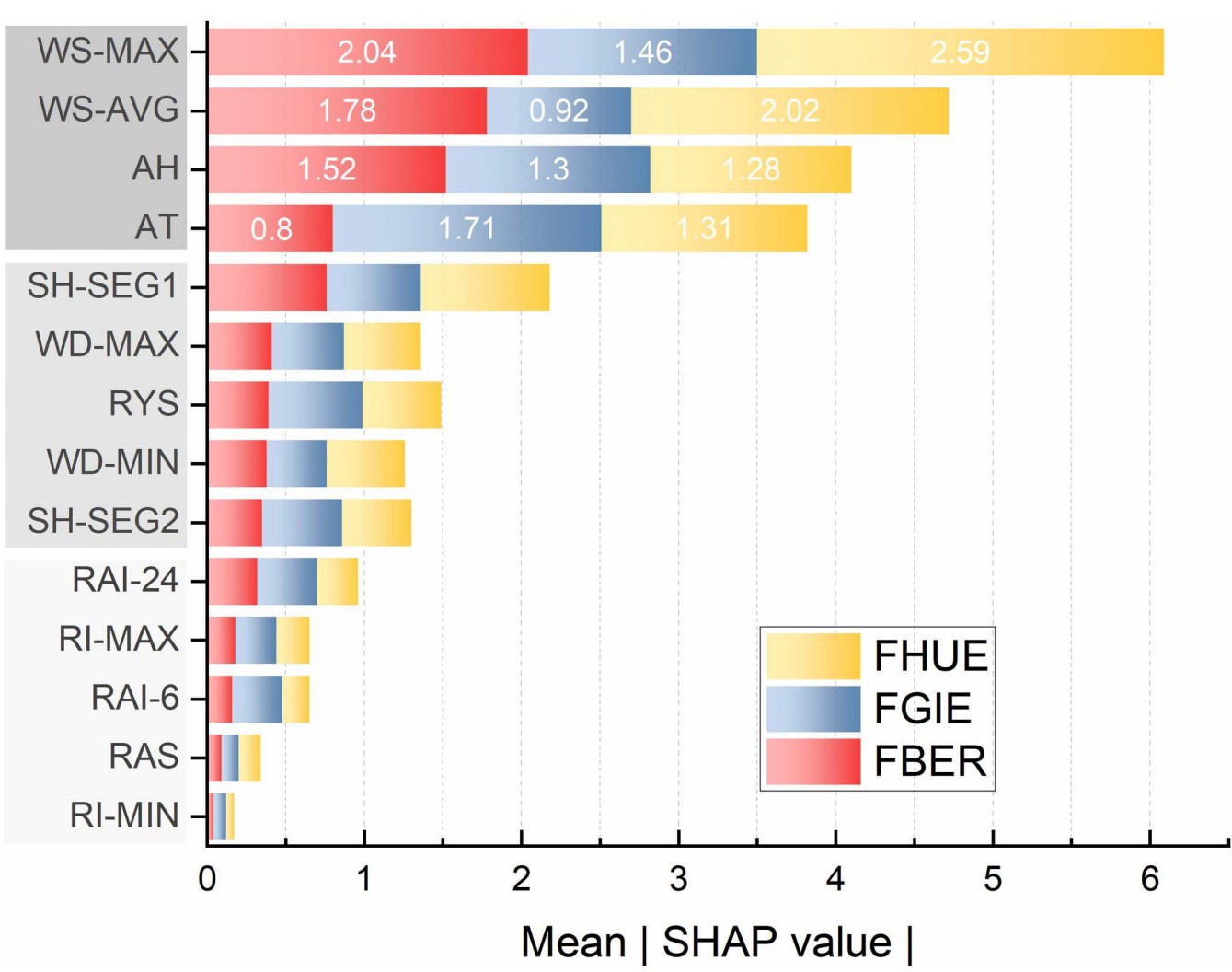

**Fig 5. Feature importance ranking based on SHAP values.**

|SHAP value|) across all samples. Fig 5 illustrates the SHAP analysis results for the three meteorological station datasets, where the bar charts, represented in different colors, correspond to the data from each station.

In Fig 5, the x-axis is essentially the average magnitude change in model output when a feature is integrated out of the model. The features are ordered by the average absolute sum value of their effect magnitudes on the model. The Y-axis indicates the feature names in order of importance from top to bottom.

It can be inferred that different meteorological features contribute variably to the drift snow prediction model, with specific features contributing significantly more than others. For instance, in the FBER and FHUE datasets, WS-MAX (maximum wind speed) dominated other features with SHAP values of 2.04 and 2.59, respectively. In contrast, for the FGIE dataset, AT (air temperature) emerged as the most critical predictor with a SHAP value of 1.71.

Additionally, the overall analysis reveals that the feature importance rankings are largely consistent across different datasets. The features that have the most significant impact on the results are WS-MAX, WS-AVG (average wind speed), AH (relative humidity), and AT. Following these in the second tier are snow height, annual precipitation, and wind direction. The least influential features are a series of parameters related to precipitation intensity.

In addition to ranking global feature importance, SHAP also provides individual explanations for each sample in the model. Fig 6 is a SHAP Summary Plot, offering insights into how the contribution of an individual feature to the model output is influenced by its value. Each point on the summary plot represents a SHAP value for a specific feature and instance. The position on the y-axis is determined by the feature, while the x-axis reflects the SHAP value. Positive SHAP values indicate that the feature contributes to Class 1 (wind-blown snow events), whereas negative SHAP values contribute to Class 0 (no-blown snow events). The color gradient represents the feature's value, ranging from low to high, for example, showing variations in air temperature or wind speed values.

Fig 6a–c present data from different meteorological stations. It can be observed that WS-MAX, WS-AVG, AH, and AT exhibit significant contributions across all three stations. The distribution of points for wind speed features shows a clear trend. For instance, WS-MAX and WS-AVG generally have a positive impact on the model output, with their SHAP values

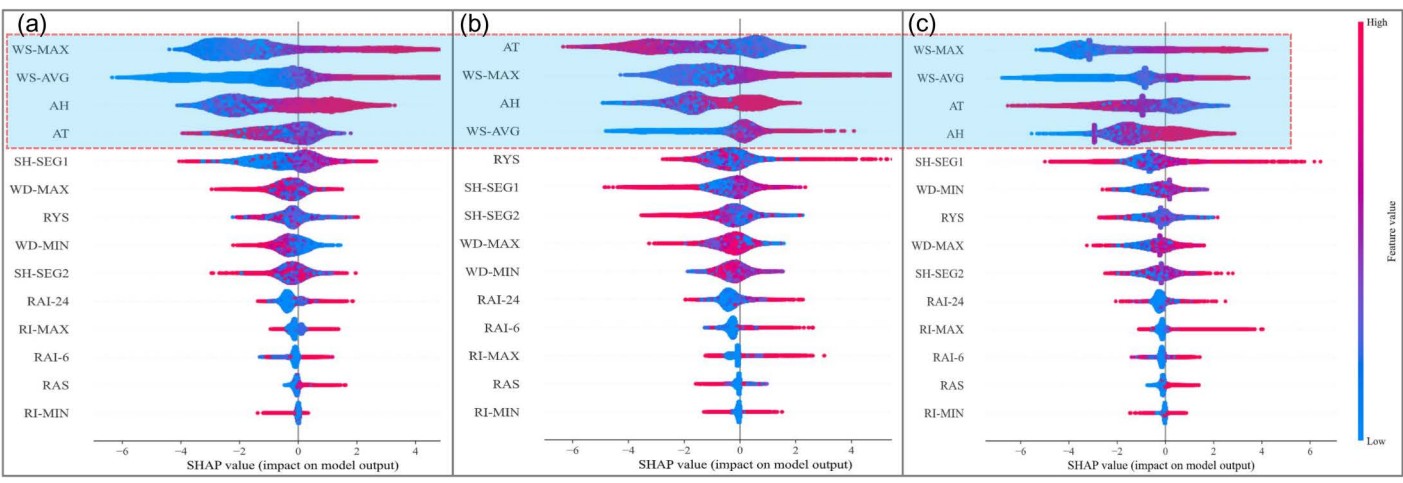

**Fig 6. SHAP Summary Plot; (a) FBER; (b) FGIE; (c) FHUE.**

positively correlated with wind speed. In other words, higher wind speeds contribute more to Class 1. Additionally, the minimum SHAP value for WS-AVG is noticeably lower than that of WS-MAX, indicating that for predicting Class 0, WS-AVG plays a more significant role than WS-MAX.

Further analysis of the relationship between SHAP values and feature values reveals that AH and AT do not exhibit as clear a trend as wind speed, as they display more intermixed blue and red points. However, some valuable patterns can still be observed. For instance, higher AH significantly contributes to Class 1. Additionally, there is a negative correlation between air temperature and the probability of Class 1. This means that higher temperatures contribute more to the no-blown snow event, whereas lower temperatures have a more pronounced contribution to the wind-blown snow event.

Since the SHAP Summary Plot only reflects the overall magnitude of feature contributions without showing the specific feature values, to gain a deeper understanding of feature importance and their contribution to the model's predictions, SHAP force plots were generated using SHAP values from the XGBoost model. Force plots are a powerful tool for visualizing the contribution of individual features to the final prediction for a single instance. To explore the relative importance of different features in predicting snowdrift, several representative cases were selected, with the results shown in Fig 7.

In Fig 7, four representative prediction samples are shown, and the prediction results are consistent with the actual outcomes. The vertical axis represents the individual features, sorted by their importance, along with their corresponding values. The horizontal axis shows the predicted probability of Class 1, expressed in log odds, which must be transformed using the following equation to obtain the final probability.

$$P\left(Class\ 1\right) = \frac{1}{1 + e^{-\left(Log\ odds\right)}} \tag{8}$$

The default decision threshold for the final probability is 0.5, meaning that a value greater than 0.5 results in a prediction of Class 1, otherwise it is predicted as Class 0. In Fig 7, $E[f(X)]$=-0.302 represents the model's output without any feature contributions, which corresponds to $E[P(Class1)]$=0.425. The value $f(x)$ is the final log-odds after summing the effects of all features. The transformed probabilities of Class 1, denoted as $P$(Class 1), for samples (a), (b), (c), and (d) are 0.000219, 0.00622, 0.0142, and 0.9992, respectively.

A closer analysis of the feature values reveals that, in Fig 7a, where the temperature is 7.5°C, humidity is 29.9%, and maximum wind speed is 12.2 m/s, most of the feature contributions push the prediction of Class 1 to a lower probability. In contrast, in Fig 7d, where the temperature is -4.7°C, humidity is 99.2%, and maximum wind speed is 25.2 m/s, most feature contributions push the prediction towards a higher probability for Class 1. This indicates that for wind-blown snow to occur, certain high-impact conditions need to be met, such as low temperatures, high humidity, and high wind speed. If one or more of these conditions are not satisfied, as in Fig 7b and 7c, it is unlikely that wind-blown snow will occur.

## 4.3  Feature dependency analysis

This section explores the role of various features in the prediction process to quantify the dependency between features and model outputs. Feature Dependency Analysis, an essential interpretability method, is introduced to reveal how input features influence the model's predictive outcomes. Fig 8 shows the univariate Partial Dependence Plots (PDP) for WS-MAX, WS-AVG, AT and AH with respect to Class 1. The core of this analysis involves varying the values of feature $x_j$ while holding other features constant and calculating the model's predicted

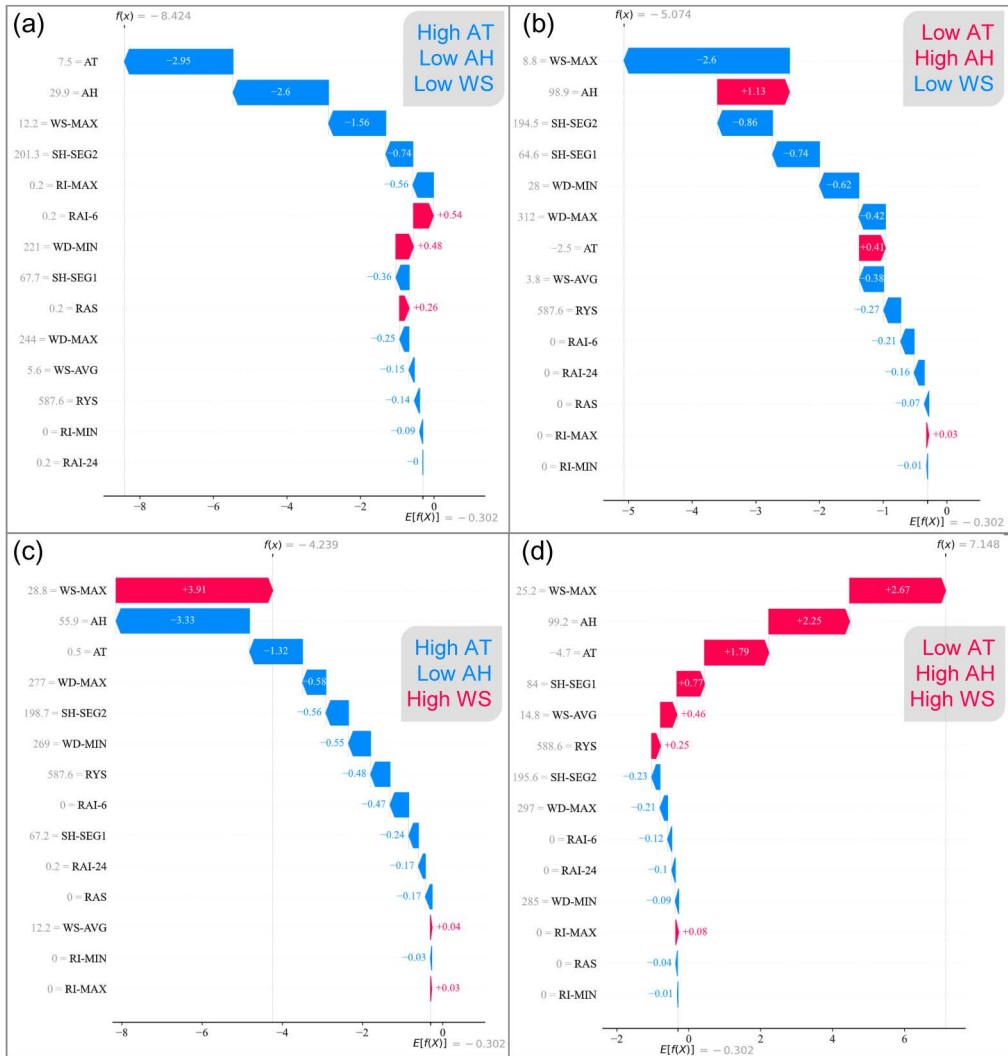

**Fig 7..** SHAP force plot showing feature impacts for individual samples: (a), (b), (c) no-blown snow event, and (d) wind-blown snow event.

output using the Partial Dependence Function (PDF) at each value. The x-axis represents the range of the individual feature values in the dataset, while the y-axis displays the average predicted output of the model for each corresponding x-axis value, indicating the probability of snow drift occurrence.

From Fig 8, it can be observed that despite the influence of the different data locations from the three meteorological stations (FBER, FGIE, FHUE), the impact of each feature on the model output remains consistent across different datasets. The differences are only marginal in terms of the absolute output values (Partial dependence). This is also in agreement with the results provided by SHAP values, where WS-MAX, WS-AVG, and AH show a significant positive correlation with the model output for Class 1, while AT exhibits a clear negative correlation.

Through an analysis of the mean curve in the one-way PDP presented in Fig 8a, it is evident that when WS-MAX exceeds 5 m/s, there is a linear relationship between WS-MAX and the Partial Dependence, indicating that as the maximum hourly wind speed at the site

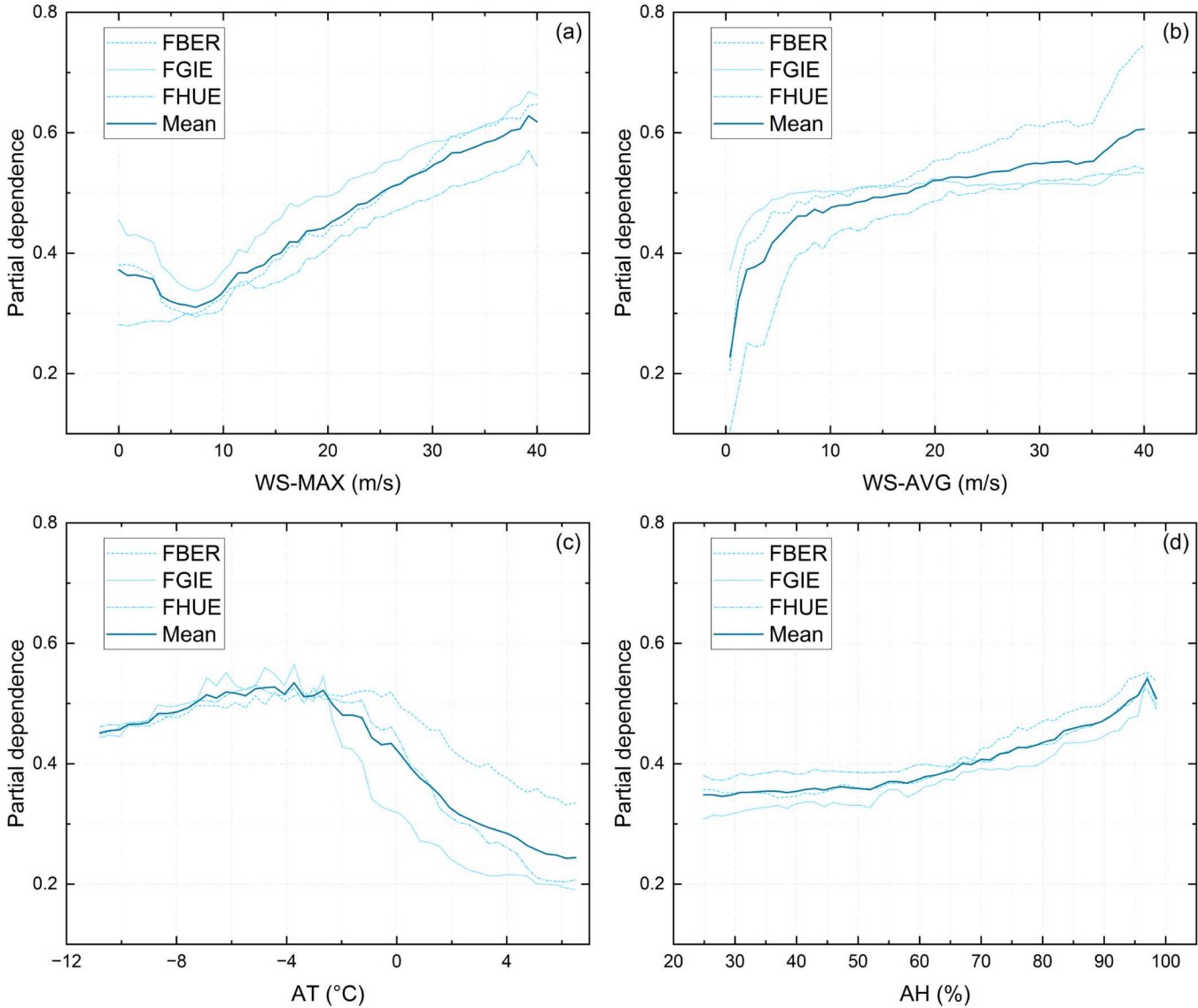

**Fig 8. One-way partial dependence plots (PDP) of (a) WS-MAX, (b) WS-AVG, (c)AT, (d) AH.**

increases, the probability of windblown snow occurrence proportionally rises. However, an anomalous upward trend is observed in the low wind speed range (with WS-MAX approaching 0 m/s) in Fig 8a. This non-linearity can primarily be attributed to the characteristics of the instrumentation. Specifically, the FlowCapt sensor may misclassify snowfall or minor air disturbances as windblown snow events under near-zero wind speed conditions (false positives). Additionally, in certain instances, actual windblown snow events might occur at very low wind speeds that fall below the detection threshold of the anemometer, leading to samples where snow flux is detected despite a recorded WS-MAX of 0 m/s.

Despite this anomaly, its impact on model predictions is minimal. The low wind speed range constitutes only a small portion of the overall dataset, and the final predictions are determined by the combined influence of all features rather than a single segment of one feature. Furthermore, the confusion matrix confirms that the model performs well overall,

suggesting that this localized non-linear effect does not significantly impair prediction accuracy. In reality, windblown snow primarily occurs at higher wind speeds, and the snow flux detected under low wind speed conditions does not reflect actual windblown snow events. Therefore, while the anomalous trend arises due to the high sensitivity of the instrument, it has negligible influence on the model's reliability and predictive capability.

In Fig 8b, it is observed that similar to WS-MAX, an increase in WS-AVG also leads to a rise in the probability of wind-blown snow events. Nevertheless, when WS-AVG exceeds approximately 10 m/s, the growth rate of the Partial Dependence curve slows considerably. This indicates that while increases in average wind speed play a significant role in predicting windblown snow events within a certain range, their influence diminishes once wind speed surpasses a critical threshold. This phenomenon can be attributed to the fact that, at wind speeds above 10 m/s, the conditions necessary to initiate and sustain blowing snow are already well met, and further increases in wind speed contribute less significantly to the probability of occurrence. At this stage, other environmental factors, such as air temperature and humidity, start to play a more prominent role in influencing the formation and persistence of windblown snow. Thus, although higher WS-AVG values still promote blowing snow events, their marginal contribution decreases as wind speed increases, highlighting the interplay of multiple factors in shaping these events.

The relationship between air temperature (AT) and the occurrence of windblown snow is more complex (Fig 8c). When AT is below -3°C, the Partial Dependence remains relatively stable around 0.5. However, as AT increases above -3°C, the influence of temperature on windblown snow becomes negatively correlated, with the Partial Dependence dropping sharply. It is also noteworthy that in the range of AT> -3°C, data from the three stations exhibit greater dispersion compared to the lower temperature range. This indicates that lower temperatures are strongly associated with the occurrence of windblown snow events, whereas higher temperatures are closely linked to a reduction in such events.

Compared to wind speed (WS-MAX, WS-AVG), the positive correlation between relative humidity (AH) and the occurrence of windblown snow is more gradual and stable. As shown in Fig 8d. When AH is below 60%, changes in humidity have little to no significant impact on the Partial Dependence, indicating that under relatively dry conditions, humidity contributes minimally to the occurrence of windblown snow. However, as AH exceeds 60%, the probability of windblown snow gradually increases, demonstrating a steady positive correlation. In particular, under high humidity conditions (approaching 100%), the model's predicted probability for windblown snow reaches its maximum value.

It is important to clarify that the conclusion of this study—that wind-blown snow is more likely to occur under high humidity conditions—refers to relative humidity (AH) in the air. Previous research has suggested that the threshold wind speed for wet snow is significantly higher than that for dry snow, meaning that wet snow requires stronger winds to be mobilized (Li & Pomeroy, 1997). The "humidity" in wet snow refers to the water content within the snowpack, rather than the water vapor content in the air. These two concepts of humidity are fundamentally different and should not be conflated. The relationship between air humidity and snowpack moisture content is not yet fully understood and may involve complex interactions. Further research is needed to explore these dynamics. This study offers a new perspective by highlighting that air humidity can serve as an important reference condition for detecting wind-blown snow events.

PDP can be extended to analyze the interaction effects between multiple features. While One-Way PDPs only show how an individual feature independently influences the prediction results, in reality, there are often complex interactions between features. Ignoring these interactions may result in an incomplete understanding of model behavior. In contrast to

One-Way PDP, two-feature PDP explores all possible values of two features ($x_j$ and $x_k$) while keeping other features fixed, and uses the Partial Dependence Function (PDF) to calculate the model's prediction results. Therefore, two-feature PDPs provide a clearer view of the predictive trends under different feature combinations, particularly how the joint influence of two variables on the target changes as both variables vary together. Considering the strong linear relationship between WS-MAX and the probability of wind-blown snow events, WS-MAX was chosen to represent wind speed instead of WS-AVG. Fig 9 illustrates the interaction effects between maximum wind speed (WS-MAX) and air temperature (AT) as well as relative humidity (AH).

Fig 9a presents the partial dependence relationship between air temperature (AT) and maximum wind speed (WS-MAX) on the probability of wind-blown snow events. In the region where -2°C <AT < 0°C and WS-MAX > 25 m/s, the Partial Dependence values are relatively high, indicating that the probability of windblown snow events is higher under these temperature and high wind speed conditions. Conversely, in the region where AT > 3°C and WS-MAX < 10 m/s, the Partial Dependence values are low, suggesting that the probability of windblown snow events is lower under higher temperatures and lower wind speeds.

It is also noteworthy that within the range of WS-MAX > 10 m/s, when AT approaches -3°C, the contour lines are most sparse, indicating that at this point, the probability of windblown snow events is least sensitive to changes in maximum wind speed. As the temperature increases or decreases, the contour lines become denser, particularly around AT approaching 3°C, where they are most concentrated. This suggests that when the temperature reaches extremely high or low values, the occurrence of wind-blown snow events become more sensitive to changes in wind speed. At around -3°C, maximum wind speed has the least influence on the probability of windblown snow events, while at higher or lower temperatures, the impact of maximum wind speed significantly increases.

It can be observed from Fig 9b that in the region where AH > 90% and WS-MAX > 20 m/s, the Partial Dependence values increase significantly, indicating that under conditions of high humidity and high wind speed, the probability of wind-blown snow events is elevated.

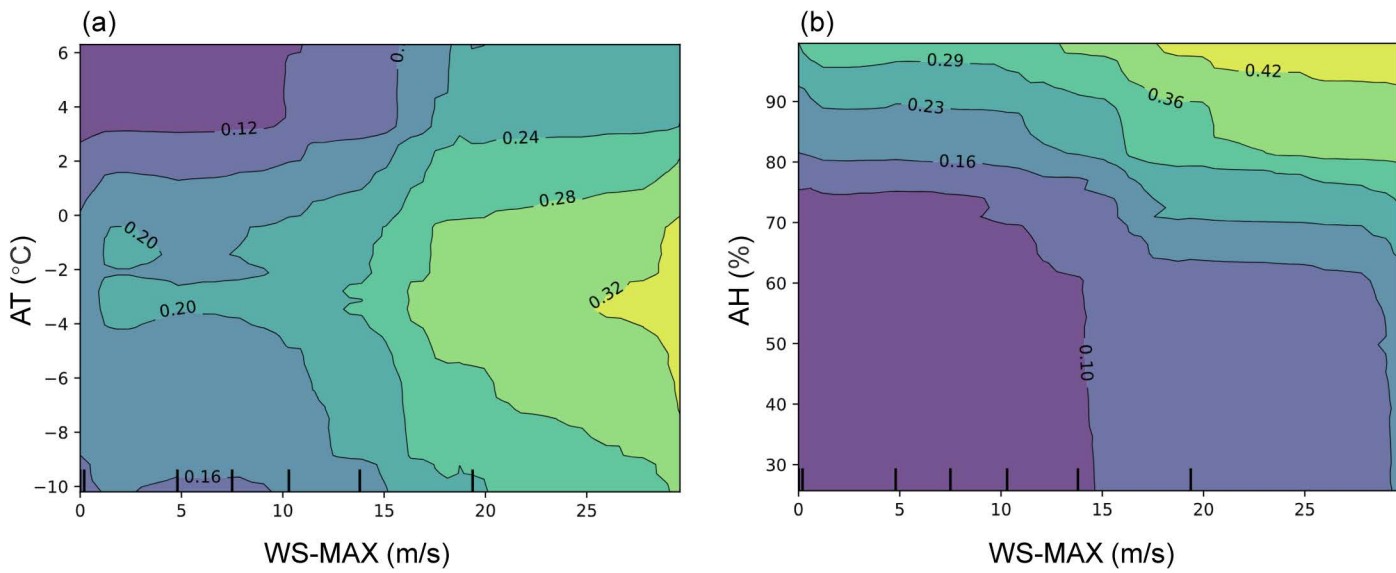

**Fig 9. The two-feature PDPs for WS-MAX with (a) AT and (b) AH.**

Conversely, in the area where AH < 70% and WS-MAX < 15 m/s, the Partial Dependence values are low, suggesting that the probability of windblown snow events is reduced under conditions of low humidity and low wind speed.

When WS-MAX close to 15 m/s, the contour lines suddenly become dense, particularly within the humidity range of 50% to 80%. This indicates that at this wind speed, the occurrence of wind-blown snow events is most sensitive to changes in humidity. In contrast, when AH < 60%, the contour lines remain relatively flat regardless of wind speed, suggesting that in the low humidity range, windblown snow events are highly insensitive to variations in wind speed.

## 5  Discussion and conclusion

Accurate prediction of blowing snow is crucial within numerical models that incorporate drifting snow. However, it also faces several challenges, including the influence of multiple complex meteorological factors and the difficulties associated with measuring the threshold wind speed under real-world conditions. In the context of rapidly advancing data analysis techniques and machine learning, recent advancements in interpretable machine learning not only significantly enhance the accuracy of drifting snow predictions but also enable the precise quantification of the impact of meteorological elements on these predictions.

In this study, field meteorological and snow flux monitoring data were utilized to train an XGBoost model for snow drift event detection. The training dataset included data from three different weather stations in the Alps region. After performing five-fold cross-validation, the performance of the XGBoost model was compared with that of other machine learning models on this dataset (e.g., SVM, RF, and GBDT), revealing that XGBoost achieved the best results. Specifically, the model exhibited a precision of over 0.94 and a recall ranging from 0.92 to 0.97 for Class 0 (no snow drift events). For Class 1 (snow drift events), the precision ranged from 0.77 to 0.80, while the recall was between 0.84 and 0.87.

The global explainability component of the SHAP framework highlights the relative importance of various meteorological factors influencing blowing snow. In the results from different weather stations in this study, the factors with the greatest impact on snow drift events were found to be WS-MAX (maximum hourly wind speed), WS-AVG (average hourly wind speed), AT (average hourly temperature), and AH (average humidity). Following these, wind direction, snow depth, and annual precipitation were noted, while the factor with the least impact was hourly precipitation intensity.

Additionally, Partial Dependence Plots (PDP) and their extended forms were utilized to explore the quantitative relationship between the predicted probabilities of snow drift events and input meteorological features in the XGBoost model. An increase in WS-MAX is linearly positively correlated with the probability of snow drift occurrence. In contrast, WS-AVG shows a diminishing growth rate beyond 10 m/s, indicating that increases in average wind speed have a significant impact on snow drift predictions only within a certain range. The relationship between temperature (AT) and blowing snow occurrence is notably strong when AT is below -3°C, while a negative correlation is observed when AT exceeds -3°C. Relative humidity (AH) has minimal impact on snow drift below 60%, but becomes positively correlated with snow drift probabilities above this threshold, particularly reaching maximum probability as humidity approaches 100%.

The two-parameters PDP reveals the interactive effects of temperature (AT), maximum wind speed (WS-MAX), and relative humidity (AH) on the probability of snow drift events. Specifically, a higher probability of occurrence is observed when -2°C <AT < 0°C and WS-MAX > 25 m/s, while lower probabilities are found in regions where AT > 3°C and WS-MAX < 10 m/s. Notably, when AT approaches -3°C, variations in wind speed have the

least impact on snow drift events; however, this influence significantly increases as temperature rises or falls. Additionally, when AH > 90% and WS-MAX > 20 m/s, the probability of occurrence markedly increases, whereas low humidity (AH < 70%) and low wind speed (WS-MAX < 15 m/s) lead to a reduction in probability. Overall, in conditions of low humidity, wind speed has a minimal effect on the occurrence of snow drift events.

This study has achieved efficient prediction of snow drift events and exploration of feature impacts through machine learning methods, providing new insights and tools for research on snow drift disasters. However, limitations exist, primarily due to data constraints; for instance, features such as topography and snow layer temperature were not included in the analysis, which may affect the comprehensiveness of the results. Moreover, while the proposed prediction model demonstrates strong predictive capabilities within the range of historical meteorological data, its applicability to future scenarios, particularly under the influence of climate change, warrants further discussion. Climate change may lead to substantial shifts in key factors such as temperature, precipitation patterns, and wind dynamics, which could alter the occurrence and characteristics of snow drift events. These changes may reduce the reliability of the model if it is applied without adaptation to evolving environmental conditions. To ensure its continued applicability, periodic updates using newly collected datasets that reflect ongoing climatic trends are essential. Additionally, validating the model with contemporary observations and incorporating predictive climate models could enhance its robustness and adaptability to future scenarios.

Future research should prioritize data quality control to improve the precision of meteorological measurements, as well as expand the scope of analysis by including additional variables such as topography, snow density, and snow layer temperature. Enhancing model accuracy through the optimization of algorithms to account for more complex environmental interactions is another critical direction. Finally, data enrichment by incorporating diverse datasets from various meteorological conditions will improve the model's generalization capabilities. These efforts will ensure that the model remains relevant and reliable, providing an improved predictive capability for snow drift events.

## Supporting information

**S1 Fig. Observed wind-blown snow events (gray) and non-wind-blown snow events (white) in the validation set (FBER), with each cell representing one hour.**
(DOCX)

**S2 Fig. Predicted from (a) XGBoost, (b) SVM, (c) RF, and (d) GBDT models compared against the observed validation set (FBER), with white cells indicating matches and red cells indicating mismatches.**
(DOCX)

**S1 Table. Statistical characteristics of the cleaned training data.**
(DOCX)

**S2 Table. Spearman correlation coefficients between the considered parameters and wind-blown snow events.**
(DOCX)

**S1 File. Cleaned FBER site dataset.**
(XLSX)

**S2 File. Cleaned FGIE site dataset.**
(XLSX)

**S3 File. Cleaned FHUE site dataset.**
(XLSX)

**S4 File. XGBoost-SHAP model implementation for blown snow detection.**
(PY)

## Acknowledgments

We would like to extend our sincere gratitude to the scientists and engineers involved in ensuring the continuous operation of measurements from the ISAW stations. The observational data collected from these stations can be accessed through the ISAW online database at http://iav-portal.com/index.php?nav=iodmisawlist. Additionally, the cleaned dataset in S Files 1, 2, and 3 and all author-generated code in S4 File are provided in the Supporting Information to support reproducibility and further research.

## Author contributions

**Conceptualization:** Feng Wang.

**Data curation:** Sai Li.

**Formal analysis:** Xinrang Wang.

**Software:** Sai Li.

**Validation:** Xinrang Wang.

**Visualization:** Feng Wang.

**Writing – original draft:** Feng Wang.

**Writing – review & editing:** Xinrang Wang.

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
