## [Decision Letter · Decision Letter 0]

15 Dec 2024

Dear Dr. Wang,

Thank you for submitting your manuscript to PLOS ONE. After careful consideration, we feel that it has merit but does not fully meet PLOS ONE’s publication criteria as it currently stands. Therefore, we invite you to submit a revised version of the manuscript that addresses the points raised during the review process.

**ACADEMIC EDITOR:**  Based on the review assessment, the manuscript needs to be revised significantly. For each review comment, please provide a detailed response indicating exact lines of the revised version to which modificarions have been made. Finally, it is not necessaery to cite references suggested by respected reviewers, while you may use them in case they improve your work.

We look forward to receiving your revised manuscript.

Kind regards,

Majid Niazkar, Ph.D.

Academic Editor

PLOS ONE

Journal Requirements:

Reviewers' comments:

Reviewer's Responses to Questions

**Comments to the Author**

1. Is the manuscript technically sound, and do the data support the conclusions?

Reviewer #1: Yes

Reviewer #2: No

2. Has the statistical analysis been performed appropriately and rigorously?

Reviewer #1: Yes

Reviewer #2: Yes

3. Have the authors made all data underlying the findings in their manuscript fully available?

Reviewer #1: No

Reviewer #2: No

4. Is the manuscript presented in an intelligible fashion and written in standard English?

Reviewer #1: No

Reviewer #2: Yes

Reviewer #1: I here summarize these comments:

Comments:

1. The innovation of the work should be explained in a better way in the introduction section

2. Consider a better title for this study.

3. Why is the XGBoost method used? Isn't it better to use other models machine learning and compare the results? For this purpose, you can get help from the following article.

https://doi.org/10.1007/s11356-024-32228-x

4. The quality of Figure 1 should be improved. Also, the figure should have latitude and longitude.

5. Other variables can also affect the blowing of snow. You can consider different variables and then identify more effective parameters with sensitivity analysis using different methods such as Monte Carlo and use them in the machine learning method. You can get ideas from the following articles

https://doi.org/10.1016/j.coldregions.2022.103682

https://doi.org/10.1007/s11600-023-01177-3

6. The quality of Figure 5 and 6 is not good and the numbers are not visible.

7. Calculate the statistical characteristics of the data used and show that in a table.

8. Section 3.1 requires more references in the text

https://doi.org/10.1111/wej.12939

9. Using the correlation coefficient such as Pearson's correlation coefficient, show the correlation between the considered parameters with snow blowing to create a better view of the data.

10. It is better to use other evaluation criteria to evaluate the accuracy of the model. For this purpose, you can get help from the following reference

https://doi.org/10.3390/w15091650

11. Add a plot of how the observed data matches the modeled data.

12. Can the results of this study be used for the future and predict future events, such as what was done in the following study? Are these results still correct despite changes such as climate change in the future? It is better to discuss this issue in the discussion section.

https://doi.org/10.1002/joc.8130

Reviewer #2: The manuscript titled "Explainable Machine Learning Model for Blowing Snow Detection and Feature Analysis Using XGBoost-SHAP" presents a research study leveraging the XGBoost algorithm combined with SHAP (SHapley Additive exPlanations) to improve the detection and interpretation of blowing snow events. The paper is well structured and well written , however, I do have some following points to further improve the manuscript quality.

1. Line 147–154: Given the long-term availability of ISAW data, do the authors plan to make the cleaned and processed dataset publicly accessible for reproducibility and further research?

2. Line no. 224: How was the threshold of 0.1 g/m²·s for defining blowing snow events determined? Was it empirically derived or based on prior literature?

3. Line 237–240: Why was SMOTE chosen over other imbalance handling techniques such as ADASYN or Tomek Links? Were comparative tests performed?

4. Line no. 362–399: Authors need to clarify whether the model’s precision and recall for blowing snow events (Class 1) were significantly affected by temporal data dependencies, considering the 13-year dataset.

5. Can the authors elaborate on how the non-linear impact of WS-MAX at low wind speeds (below 5 m/s) was addressed? Does this anomaly significantly affect model predictions?

6. Line 483–537: While discussing the relationship between WS-AVG and blowing snow, could the diminishing impact beyond 10 m/s be attributed to hardware limitations of the FlowCapt sensor?

7. Line 558–588: In Fig. 8, the interaction of WS-MAX with AT and AH showed clear thresholds. How sensitive are these thresholds to the inclusion of additional meteorological features not considered in this study, like snow density or layer temperature?

8. Line 633–642: Considering the regional focus on the Alps, how generalizable are the model results to other geographies, such as polar regions with distinct meteorological patterns?

9. Improve the Figure no. 5 and 6 image quality

10. Authors may include a few more recent literatures on machine learning methods and XGBoost to strengthen the literature sections. Some are given below:

• doi/10.2166/hydro.2024.119/105935

• https://doi.org/10.2166/hydro.2024.292

• https://link.springer.com/article/10.1007/s11269-024-03879-9

• https://doi.org/10.1016/j.envsoft.2024.105971

• https://doi.org/10.1016/j.jhydrol.2021.126382

**Do you want your identity to be public for this peer review?** For information about this choice, including consent withdrawal, please see our Privacy Policy

Reviewer #1: No

Reviewer #2: **Yes: ** Dr Bhabani Shankar Das

---

## [Author Response · Author response to Decision Letter 1]

10 Jan 2025

Dear Reviewers and Academic Editor,

We would like to express our sincere gratitude to the reviewers and the academic editor for their valuable feedback and constructive suggestions. We have carefully reviewed each of the comments and have provided detailed responses and revisions accordingly. Your insightful recommendations have significantly enhanced the quality of our manuscript, particularly in areas such as figure formatting, selection of training parameters, presentation of training data, model evaluation, and the discussion of results, among others.

In our submission, we provide a thorough response to your comments, following the order of

Reviewer #1,

Reviewer #2,

and the Academic Editor ,

along with the corresponding revisions made to the manuscript. We hope that the revised manuscript now comprehensively addresses your concerns and meets the standards set by PLOS ONE.

Sincerely,

Feng Wang

---

## [Decision Letter · Decision Letter 1]

23 Jan 2025

Explainable Machine Learning for Predictive Modeling of Blowing Snow Detection and Meteorological Feature Assessment Using XGBoost-SHAP

PONE-D-24-48529R1

Dear Dr. Wang,

We’re pleased to inform you that your manuscript has been judged scientifically suitable for publication and will be formally accepted for publication once it meets all outstanding technical requirements.

Kind regards,

Majid Niazkar, Ph.D.

Academic Editor

PLOS ONE

Additional Editor Comments (optional):

The manuscript can be accepted for publication as it addressed the review comments.

Reviewers' comments:

Reviewer's Responses to Questions

**Comments to the Author**

Reviewer #1: All comments have been addressed

Reviewer #2: All comments have been addressed

2. Is the manuscript technically sound, and do the data support the conclusions?

Reviewer #1: Yes

Reviewer #2: Yes

3. Has the statistical analysis been performed appropriately and rigorously?

Reviewer #1: Yes

Reviewer #2: Yes

4. Have the authors made all data underlying the findings in their manuscript fully available?

Reviewer #1: Yes

Reviewer #2: Yes

5. Is the manuscript presented in an intelligible fashion and written in standard English?

Reviewer #1: Yes

Reviewer #2: (No Response)

Reviewer #1: Dear Editor

The revised manuscript addresses all the concerns raised, and I recommend it for acceptance

Reviewer #2: All the raised comments have been carefully addressed, and the document is now in an acceptable state. It can be considered for approval in its present form.

**Do you want your identity to be public for this peer review?** For information about this choice, including consent withdrawal, please see our Privacy Policy

Reviewer #1: **Yes: ** Mohammad Reza Goodarzi

Reviewer #2: **Yes: ** Dr. Bhabani Shankar Das

---

## [Editor Report · Acceptance letter]

PONE-D-24-48529R1

PLOS ONE

Dear Dr. Wang,

I'm pleased to inform you that your manuscript has been deemed suitable for publication in PLOS ONE. Congratulations! Your manuscript is now being handed over to our production team.

Kind regards,

on behalf of

Dr. Majid Niazkar

Academic Editor

PLOS ONE
